# Thermal coupling of the Indo-Pacific warm pool and Southern Ocean over the past 30,000 years

Shuai Zhang [1,2], Zhoufei Yu [3] ✉, Yue Wang [4], Xun Gong [5,6,7], Ann Holbourn[8], Fengming Chang [2], Heng Liu[1], Xuhua Cheng[1] & Tiegang Li [9,10] ✉

The role of the tropical Pacific Ocean and its linkages to the southern hemisphere during the last deglacial warming remain highly controversial. Here we explore the evolution of Pacific horizontal and vertical thermal gradients over the past 30 kyr by compiling 340 sea surface and 7 subsurface temperature records, as well as one new ocean heat content record. Our records reveal that La Niña-like conditions dominated during the deglaciation as a result of the more intense warming in the western Pacific warm pool. Both the subsurface temperature and ocean heat content in the warm pool rose earlier than the sea surface temperature, and in phase with South Pacific subsurface temperature and orbital precession, implying that heat exchange between the tropical upper water column and the extratropical Southern Ocean facilitated faster warming in the western Pacific. Our study underscores the key role of the thermal coupling between the warm pool and the Southern Ocean and its relevance for future global warming.

The last deglacial warming has been typically attributed to the retreat of icebergs over the northern high latitudes due to the increasing summer insolation[1]. However, it was also demonstrated that sea surface temperature (SST) changes in the southern hemisphere and Indo-Pacific warm pool (IPWP) preceded northern hemisphere and global ice volume variations[2–5], and that the southern hemisphere itself could induce global warming directly through its response to orbital forcing[2,6]. Furthermore, the ocean-atmospheric teleconnections between the tropical Pacific and Southern Ocean have a profound impact on global climate[6,7]. However, the role of the IPWP, the world's greatest heat engine[8], and its thermal linkages with the southern high latitudes during the last deglaciation are still ambiguous.

The El Niño-Southern Oscillation (ENSO) affects the heat distribution in the equatorial Pacific and is a major force behind the teleconnections influencing global climate[8]. This inherently unstable oceanic–atmospheric oscillation between positive (El Niño) and negative (La Niña) phases exhibits a periodicity of 2 to 8 years[9]. As a result, changes in the zonal equatorial temperature gradient across the Pacific have been used as a proxy for ENSO-like variations[10,11]. However, the ENSO evolution since the last glacial epoch is controversial owing to divergent SST changes in the equatorial Pacific under

[1]College of Oceanography, Hohai University, Nanjing 210098, China. [2]Key Laboratory of Marine Geology and Environment, Institute of Oceanology, Chinese Academy of Sciences, Qingdao 266071, China. [3]State Key Laboratory of Palaeobiology and Stratigraphy, Nanjing Institute of Geology and Palaeontology, Chinese Academy of Sciences, Nanjing 210008, China. [4]State Key Laboratory of Marine Geology, Tongji University, Shanghai 200092, China. [5]Institute for Advanced Marine Research, China University of Geosciences, Guangzhou 511455, China. [6]State Key Laboratory of Biogeology and Environmental Geology, Hubei Key Laboratory of Marine Geological Resources, China University of Geosciences, Wuhan 430074, China. [7]Shandong Provincial Key Laboratory of Computer Networks, Qilu University of Technology (Shandong Academy of Sciences), Jinan 250014, China. [8]Institute of Geosciences, Christian-Albrechts-University, Kiel D-24118, Germany. [9]Key Laboratory of Marine Geology and Metallogeny, First Institute of Oceanography, Ministry of Natural Resources, Qingdao 266061, China. [10]Laboratory for Marine Geology, Pilot National Laboratory for Marine Science and Technology (Qingdao), Qingdao 266237, China. ✉e-mail: zfyu@nigpas.ac.cn; tgli@fio.org.cn

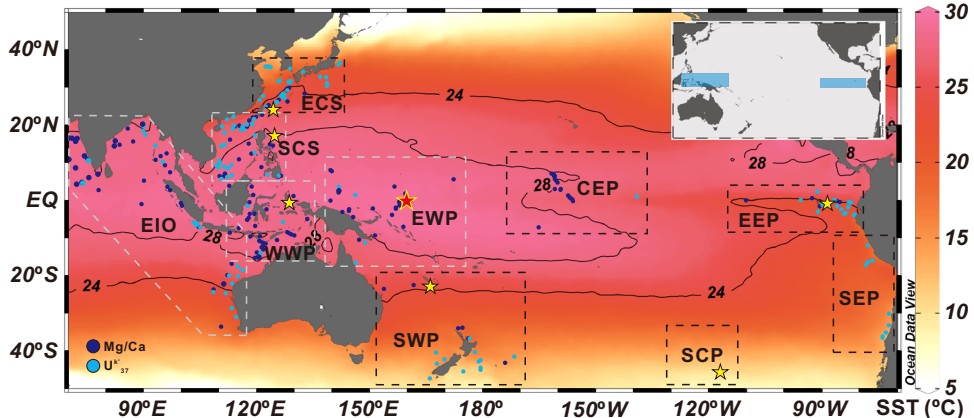

**Fig. 1 | Locations of published sea surface temperature (dots) and subsurface temperature (yellow stars) records from the Indo-Pacific region referred to in this study.** Red star marks the ocean heat content record reconstructed from core KX22-4 in this study. The base map shows modern annual mean sea surface temperature distribution drawn with Ocean Data View software[83] based on the World Ocean Atlas 2018 dataset[84]. The isothermals of 28 and 24 °C delineate the modern extents of the Indo-Pacific warm pool (IPWP) and eastern Pacific cold tongue. The dashed rectangles mark the East China Sea (ECS), South China Sea (SCS), Eastern Indian Ocean (EIO), west and east part of the western Pacific warm pool (WWP and EWP), central and eastern equatorial Pacific (CEP and EEP), southwest, south-central and southeast Pacific (SWP, SCP, and SEP), respectively. The white rectangles mark the areas within the Indo-Pacific warm pool region, all areas are classified based on cluster analysis (Supplementary Fig. 1). The insert map shows the areas considered for El Niño-Southern Oscillation-like investigations (blue boxes).

heterogeneous hydrological conditions. The last glacial maximum (LGM) SST cooling in the IPWP has a range of 2–5 °C based on geochemical proxies[10,12,13], while cooling in the eastern equatorial Pacific (EEP) cold tongue varies between 0.3 and 3.5 °C[14,15]. Some previous ENSO investigations included cores that were distant from the central western Pacific warm pool (WPWP)[16,17], or in areas close to shore that were impacted by coastal upwelling, fresh water runoff and coastal currents[13,18,19]. Other ENSO investigations were based on the meridional SST gradient between the EEP cold tongue and the northeastern Pacific warm pool, which is under debate as an indication of ENSO or seasonality assessment[20,21]. These discrepancies ultimately result in opposing assertions about the prevalence of La Niña-like[10,13,17,19,20] or El Niño-like conditions[5,11,12,14,16] in the last glacial period.

In this study, we integrated 340 published SST data sets (planktonic foraminiferal Mg/Ca and $U^{k'}_{37}$ alkenone index) spanning the last 30 kyr from the tropical and subtropical Pacific Ocean and eastern Indian Ocean (Fig. 1) to gain a comprehensive spatial and temporal overview of thermal variability: (1) We quantified the variations in the boundaries of the WPWP, the warmest region within the IPWP, by analyzing the temporal SST distribution in the IPWP. (2) We monitored ENSO-like transitions since the last glacial period by comparing SST anomaly (SSTA) stacks between the warm pool core area and the EEP cold tongue (Fig. 1). (3) We spatially elucidated the thermal linkage of the tropical Pacific to the Southern Ocean and their respective role during deglacial warming, by combining a new ocean heat content (OHC) record (based on Mg/Ca ratio of five planktonic foraminiferal species from core KX22-4 in the WPWP, Supplementary Material) and seven published subsurface temperature (subT) records in the Pacific (Fig. 1).

## Results and discussion
### ENSO-like variations of the IPWP thermal state since 30 ka
The ENSO-like variations depicted by the ΔSST between the WPWP and EEP since the last glacial period are shown in Fig. 2d. The glacial zonal ΔSST was considerably smaller (0.67 ± 0.24 °C) than the modern annual mean ΔSST (Fig. 2d), suggesting a more El Niño-like state during the last glacial. This is also supported by transient simulation (TRACE) results, which display 0.82 ± 0.27 °C less glacial cooling in the EEP than in the WPWP, in contrast to the early Holocene (5–10 ka) (Supplementary Fig. 2)[22]. This asymmetric cooling may be linked to the inhibited upwelling of subsurface waters in the cold tongue region,

ascribed to the deeper thermocline in the EEP and reduced Walker circulation[12,23]. With a 2.41 ± 0.63 °C cooling during the glacial (Supplementary Table 1), the WPWP migrated slightly toward the equator[24] (Fig. 3a, c) and its latitudinal extent was reduced by about 4° compared to the Holocene (Fig. 3d and Supplementary Figs. 3, 4), meanwhile its west boundary shrank markedly eastward to ~150°E (Fig. 3d and Supplementary Figs. 3, 4). A prevailing weaker and eastward tropical Pacific Walker circulation[25] in the glacial consequently contributed to a reduced northward heat transfer through the weakened Kuroshio Current[26] and the cooling of Indonesian seas by the weakened Indonesian Throughflow[19].

In contrast, the meridional extent of the WPWP in the early Holocene approximated from 19°N to 14°S (Fig. 3a, c), consistent with the present 28 °C isotherm (Fig. 1), and its west boundary in the early Holocene was at ~120°E (Fig. 3b, d). As evidenced by the precipitation records from lacustrine sediments[27,28] and modeling studies[29,30], a gradually increased frequency of El Niño-like events occurred over the late Holocene (Fig. 2c, d). The attenuated Walker circulation caused a decline in tropical heat export from the IPWP via the Indonesian Throughflow to the Leeuwin Current offshore western Australia from the mid to late Holocene[31]. Besides, this period was characterized by a reduction in thermocline nutrients and $CO_2$ leakage in the EEP[32,33]. Because El Niño-like conditions during the Holocene were not as pronounced as during the last glacial, this relatively mild El Niño-like tendency would have also contributed to $CO_2$ release into the atmosphere.

During the deglaciation, more intense warming in the WPWP than in the EEP resulted in a larger zonal ΔSST contrast and a more La Niña-like state (Fig. 2d, e and Supplementary Fig. 5). Results of the WPWP spatial development from TRACE reveal that the WPWP first reached 28 °C at ~14 ka (Supplementary Fig. 6) during the Bølling-Allerød Warm interval[22], and then remained above 28 °C since ~12 ka (Supplementary Fig. 6) following the Younger Dryas cold event, in phase with the precession minimum (Fig. 2d). Meanwhile, the extent of the WPWP began to expand substantially at ~12 ka when La Niña-like conditions peaked, the WPWP OHC also reach highest values (Fig. 2b, d and Fig. 3d). This thermal expansion was in line with a record from the East China Sea which showed that warm water diatom species increased since 12 ka[34].

The more intense deglacial warming in the WPWP may be attributed to the presence of a barrier layer, a feature well-recognized in

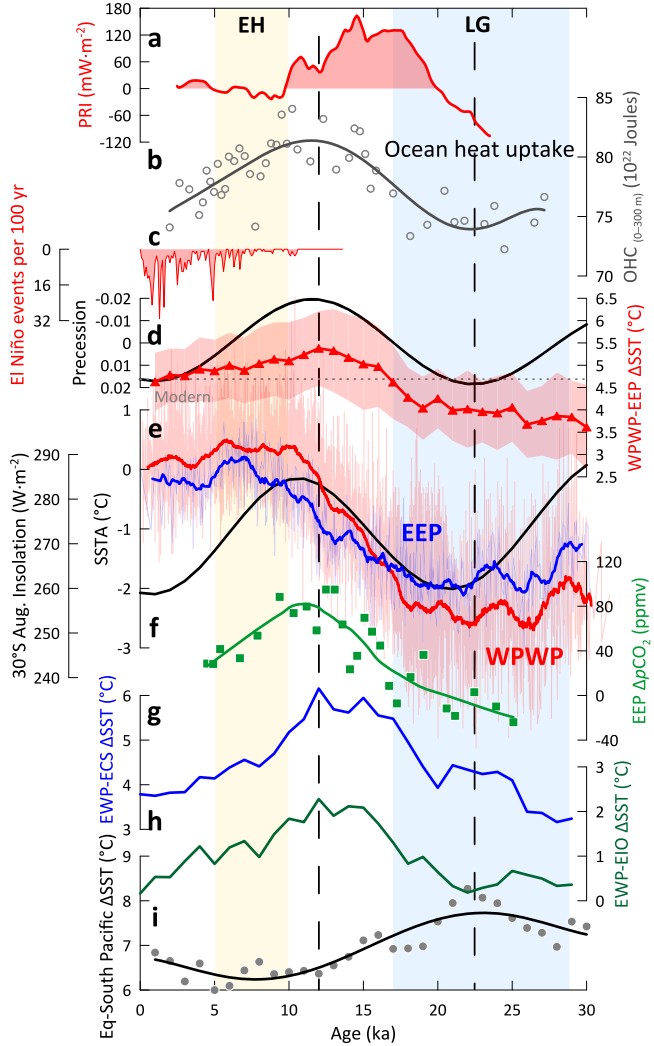

**Fig. 2 | Thermal variations of the Indo-Pacific warm pool and El Niño-Southern Oscillation-like evolution since 30 ka. a** Variations of planetary radiative imbalance (PRI)[61]. **b** 0–300 m ocean heat content (OHC) record from core KX22-4 in this study (Supplementary Fig. 17). **c** El Niño events recorded in lake Laguna Pallcacocha[27] in the eastern equatorial Pacific. **d** Annual zonal sea surface temperature difference between the central western Pacific warm pool (WPWP) and eastern equatorial Pacific (EEP) (red) and its standard deviation (red shade), the dotted horizon line represents the modern annual mean sea surface temperature difference based on the World Ocean Atlas 2018 sea surface temperature data, and black solid line denotes orbital precession[85]. **e** Annual mean sea surface temperature anomaly (SSTA) stacks in the central western Pacific warm pool (red, Mg/Ca) and eastern equatorial Pacific (blue, Mg/Ca and U$^k_{37}$, Supplementary Fig. 5) computed using a sliding rectangular window of 1 kyr width by STATNARY 1.2, as well as the late winter insolation at 30°S (black). **f** Oceanic $\Delta p CO_2$ record from Ocean Drilling Program Site 1238 in the eastern equatorial Pacific[32]. **g–i** Zonal and meridional heat transfer (sea surface temperature gradient) between the eastern part of the warm pool (EWP) and East China Sea (ECS), the eastern part of the warm pool and eastern Indian Ocean (EIO), the equatorial Pacific (Eq) and South Pacific (Supplementary Fig. 9). Yellow and blue bars indicate the early Holocene (EH) and last glacial (LG), respectively. The vertical dash lines denote the precession maximum and minimum at -22 and -12 ka, respectively.

modern observations and models[35,36]. The barrier layer forms beneath the fresh surface water and at the base of the mixed layer in the WPWP due to the vigorous atmospheric deep convection there[35]. It can effectively restrict the entrainment of the cold thermocline water to the mixed layer, favoring heat built up at the sea surface[35,37–39]. Thus, the barrier layer along the equatorial Pacific is intimately linked to ENSO activity[36,39–41]. During the deglaciation, the strong atmospheric

convection in the WPWP[10,42] benefited the development of the barrier layer and contributed to the buildup of surface heat. To explore this further, we examined the Mg/Ca-SST and residual seawater $\delta^{18}O$ ($\delta^{18}O_{sw-iv}$) data of *Globigerinoides ruber* and *Trilobatus sacculifer* from core KX22-4 in the east part of the warm pool (Supplementary Fig. 7). The SST contrast between these two species somewhat decreased in the deglaciation (remained about 0.5 °C), which means that *T. sacculifer* inhabits the mixed layer as *G. ruber*. While the $\delta^{18}O_{sw-iv}$ contrast between them dramatically rose due to increased $\delta^{18}O_{sw-iv}$ of *T. sacculifer*, which implies that *T. sacculifer* resides in waters that are much saltier than *G. ruber*, probably within the barrier layer that contributed to the prevalence of La Niña-like conditions[37,38,43].

### Deglacial coupling between the IPWP and Southern Ocean

Both atmospheric and oceanic processes are involved in the teleconnections between the tropical Pacific and Southern Ocean[44,45]. Data reanalysis and modeling experiments show that the Southern Ocean heat uptake alters the meridional circulation and is accompanied by anomalous easterlies, resulting in a zonally asymmetric La Niña-like pattern of SST change through atmospheric process[42,46,47]. In this study, we focus on the oceanic tunnel between the IPWP and the Southern Ocean. The Southern Hemisphere supplies up to 70% to the water masses in the equatorial undercurrent[48] and serves as an external thermal source to discharge/recharge the tropical ocean[49]. According to observations and simulations, the shallow meridional overturning circulation/subtropical overturning cells connect the extratropical and tropical oceans. These processes include the subduction of subtropical water, equatorward geostrophic flow in the thermocline to the equatorial Pacific either through its interior or western boundary pathway, and upwelling at the equator[48,50–53] (Fig. 4g). Subduction of South Pacific waters includes Subtropical Mode Water, Subantarctic Mode Water and Antarctic Intermediate Water[51,54]. Subantarctic Mode Water and Antarctic Intermediate Water formed in the Southern Ocean are transported northward into the adjacent subtropical gyres[55].

Around 75% of the heat absorbed by the ocean over the historical period has come from the Southern Ocean, north of the Antarctic Circumpolar Current[56,57], and heat is preferentially stored where surface waters are subducted to the north[58]. Modern heat gained by the Southern Ocean south of 30°S[57] is carried through the subpolar-subtropical gyre toward the equator and plays an important role in ventilating the equatorial thermocline, on which ENSO variability heavily depends[49,50,59,60]. The overturning circulation causes the eastern edge of the warm pool to move westward during La Niña events. This alters the thermocline structure and surface stratification, which in turn causes the thick barrier layer to move back and forth along the equator and is thought to have an impact on ENSO evolution[53]. When more heat entered the ocean with increased insolation[61] as shown by the raised planetary radiative imbalance in the late last glacial epoch (Fig. 2a), a weaker Atlantic Meridional Overturning Circulation in a cold climate period would cause greater heat to accumulate in the Southern Ocean by impeding heat transfer into the boreal high latitudes[62]. As the excessive heat in the southern hemisphere was transported equatorward into the interior of the ocean by subduction[58], subT of the eastern part of the WPWP rose earlier than the SST at -22 ka (Fig. 4d). This is consistent with a previous study, which showed that subT warming began about 4000 years before the deglacial $p CO_2$ and SST rise in the IPWP[63]. An earlier increase in WPWP subT and a decrease in the vertical temperature gradient also occurred in the south Pacific (Fig. 4e, f), in contrast to the delayed subT warming and the increase in the vertical temperature gradient in the East China Sea and eastern Indian Ocean regions (Fig. 4a, b). This earlier increase probably supports a thermocline linkage between the WPWP and Southern Ocean during the deglaciation. Furthermore, the temperatures of the EEP cold tongue, which is fed by the equatorial undercurrent from the western Pacific, also rose earlier[64,65] (Fig. 4c). Evidences from $\Delta^{14}C$ also indicated a

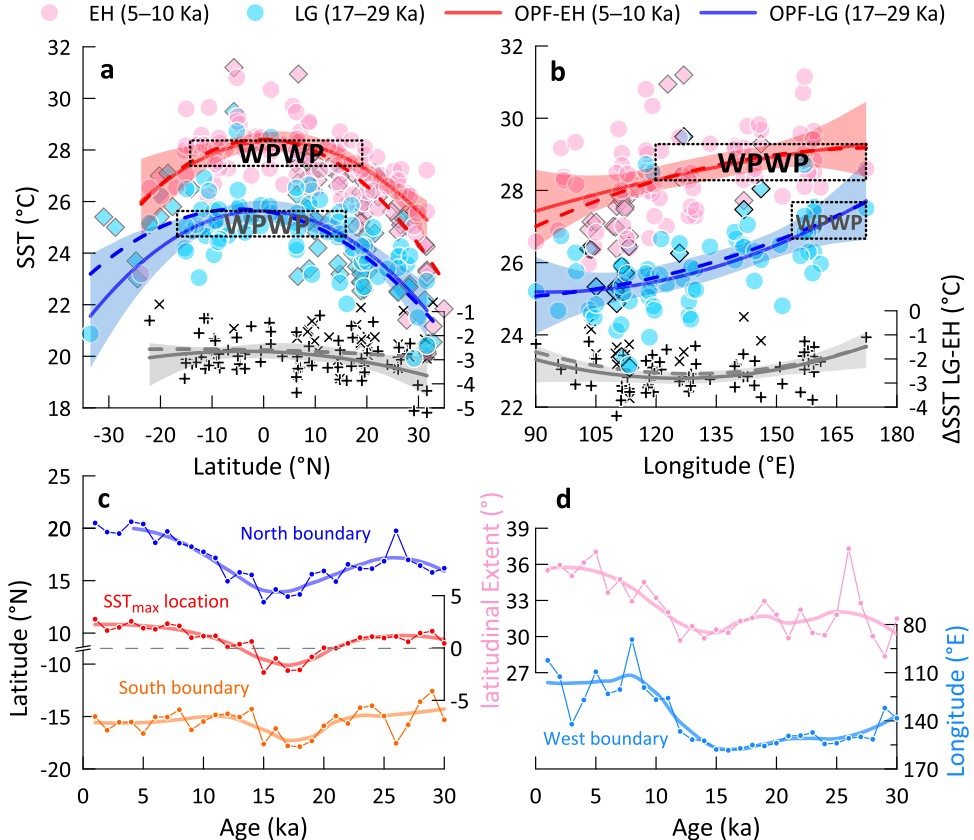

**Fig. 3 | Variations in the spatial extent of the western Pacific warm pool since 30 ka. a** Comparisons of meridional and **b** zonal variations in the extent of the western Pacific warm pool (range of top 1 °C) in the last glacial (blue) and early Holocene (red), based on sea surface temperature distributions. Trend lines are orthogonal polynomial fittings (OPF) based on Mg/Ca (solid) and mixed temperature proxies (Mg/Ca and $U^k_{37}$, dashed), with 90% confidence (shadows). Gray lines are discrepancies between the last glacial and early Holocene. Mg/Ca results are shown by circles and plus signs, while $U^k_{37}$ results are represented by diamonds and crosses. The spatial extent of the western Pacific warm pool marked by dotted boxes is based on sea surface temperature data from Mg/Ca, which are more uniformly distributed compared to sea surface temperature based on $U^k_{37}$ (Supplementary Fig. 10). Mg/Ca based sea surface temperature distribution did not differ much from annual sea surface temperature distribution ($U^k_{37}$ included) in the western Pacific warm pool region with no discernible seasonal variability (−0.07 ± 0.76 °C), divergence was more marked at higher latitude owing to the data weighted towards sea surface temperature based on $U^k_{37}$ (Supplementary Fig. 7). Please refer to the supplementary material for a discussion of the seasonality of proxies. **c** Meridional displacements of the north and south boundaries and location of maximum sea surface temperature within the western Pacific warm pool. **d** Variations in the latitudinal extent and west boundary of the western Pacific warm pool.

Southern Ocean source for the waters in the EEP and even eastern North Pacific (23.5°N) via the equatorial undercurrent[66-68] during the deglaciation. The results of TRACE showed that the upper water column of the WPWP absorbed heat from the south and lower waters, and released mainly to the north (Supplementary Fig. 8) in the deglacial warming, and the SST gradient between the equatorial and south Pacific reduced (Fig. 2i).

A continuous subduction of the extratropical warm anomaly to the WPWP thermocline is hypothesized to affect ENSO-like transitions since 360 ka, paced by orbital precession[69]. The results of earlier modeling simulations have shown that orbital precession exerts a major impact on ENSO-like processes through the strength of ocean-atmosphere feedbacks[70,71]. Precessional forcing on the subtropical South Pacific affects the stratification of the tropical Pacific, altering the strength of ocean upwelling and the thermocline tilt to the surface wind anomalies[70]. When the precessional parameter decreased in the deglaciation (Fig. 2d), the overturning circulation that mainly formed in winter by intense vertical mixing[72] was warmed due to the rising insolation in austral winter (Fig. 2e), and this warmer signal was further transferred to the western Pacific[69]. Preceding thermocline anomalies in the western and central deep tropical Pacific contribute to the recharge phase of ENSO[63,73], and modern observations also indicate that the change in the WPWP OHC is tightly related to ENSO activity[74]. During the deglaciation, rise in WPWP subT, OHC, as well as ENSO-like

variations were quite correlated with the decrease in the precession parameter (Fig. 2b, d and Fig. 4d).

We argue that the rapid warming in the subsurface layer and reduced vertical temperature gradient in the WPWP also favored the intense deglacial SST increase and a La Niña-like mode. The intense SST rise in the WPWP increased the meridional and zonal temperature gradients (Fig. 2g, h) and reinvigorated heat transport to the high latitudes[5,63]. Ekman transport converged surface warm waters to the subtropics[75], lowering the ΔSST between the WPWP and the East China Sea (Fig. 2f). The recharge of the equatorial warm pool supplied the Indonesian Throughflow and the Equatorial Undercurrent, thereby feeding the downstream surface waters of the cold tongue in the eastern Pacific[76]. The enhanced Walker circulation, in turn, facilitated the buildup of warm water westward by intensifying the equatorial easterly winds[77]. Under the La Niña-like state with more heat gathered in the WPWP, the Pacific thermocline tilted and the shoaling of thermocline in the EEP, currently the largest source of $CO_2$[78,79], facilitated the outgassing of $CO_2$ (Fig. 2e)[11,32,33,80], further accelerating the global warming.

In this study, we compiled 340 SST and 7 subT published records from the Pacific and eastern Indian Ocean, as well as a new OHC record from the WPWP, to elucidate the thermal evolution of the IPWP region and its coupling the Southern Ocean during the last deglaciation. The zonal SST gradient in the tropical Pacific Ocean was reduced and El Niño-like conditions dominated in the last glacial, as the WPWP

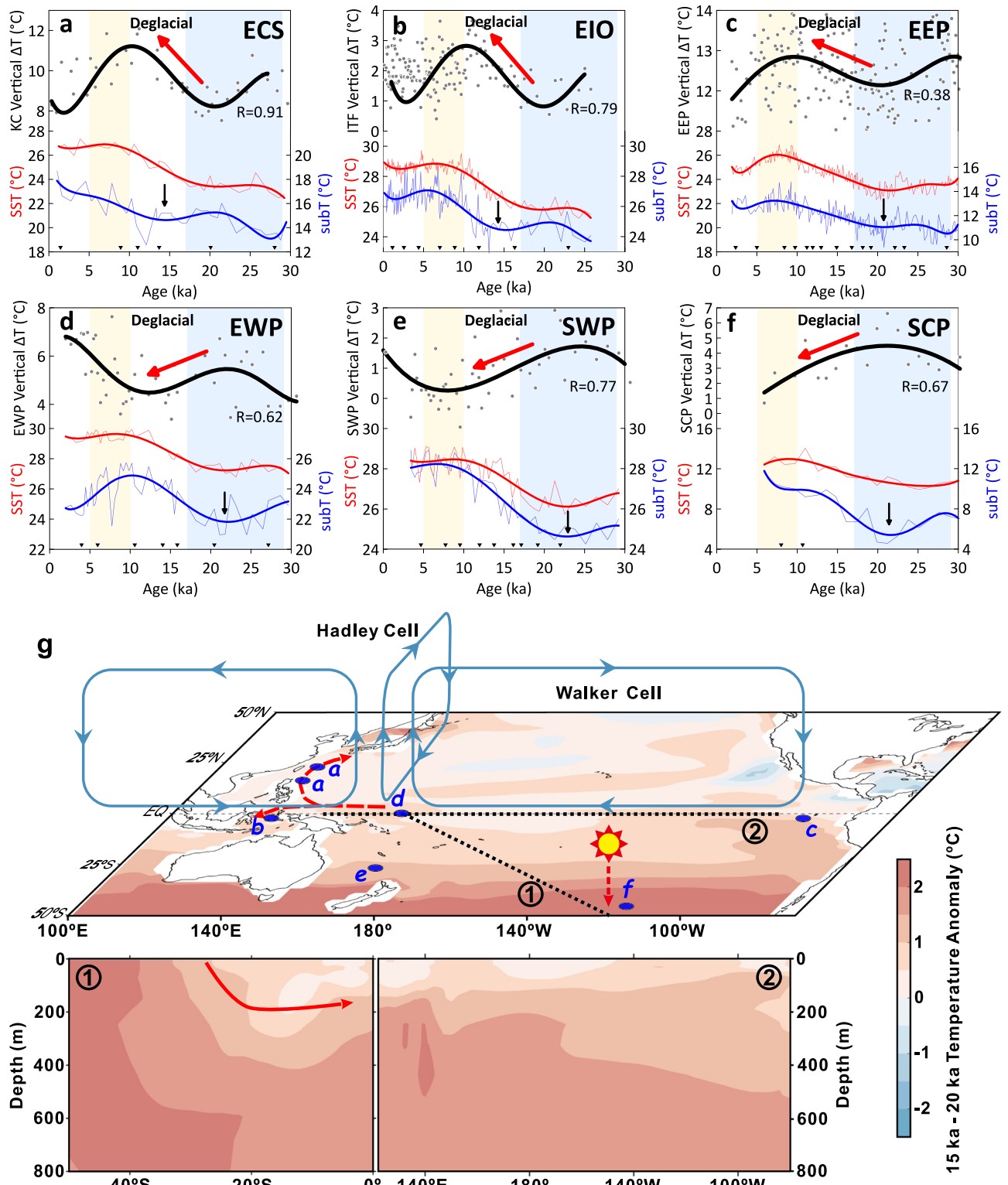

**Fig. 4 | Hypothesized spatial heat transfer process in the Pacific during the last deglaciation. a–f** Variations of sea surface temperature (SST), subsurface temperature (subT) and their vertical gradient since 30 ka along the path of the Kuroshio Current (KC) in the East China Sea (ECS)[17,86], along the path of the Indonesian Throughflow (mirror the variation in the eastern Indian Ocean: EIO)[63], in the eastern equatorial Pacific (EEP)[64], in the east part of western Pacific warm pool (EWP), in the southwest Pacific (SWP)[87–89] and south-central Pacific (SCP), respectively, as shown on the map. **g** The bold sea surface temperature, subsurface temperature and their temperature difference (ΔT) lines are polynomial fitting results for the raw data (gray), the *P* value of the polynomial fitting for ΔT is 0.0003 for East China Sea and <0.0001 for all the other regions. The inverted triangles indicate [14]C-dated points. The black downward arrows mark the start of subsurface temperature warming. **g** TRACE modeled temperature anomaly in the deglaciation relative to glacial period (15 minus 20 ka). The red dashed arrows in the upper panel indicate surface heat transfer from the western Pacific warm pool. The sun marked warming in the south Pacific subduction area. Two vertical sections marked by dotted lines are shown in the lower panel, a hypothesized oceanic tunnel of heat transfer (red arrow) between the Southern Ocean and equatorial Pacific is sketched out.

contracted towards the equator and the east. During the deglaciation, La Niña-like conditions prevailed as a result of the intense warming in the WPWP, which may have been related to the development of the barrier layer within the mixed layer, favoring heat built up at the sea surface. Both the subT and OHC in the warm pool rose earlier than the SST, and in phase with South Pacific subTs and orbital precession, implying that heat exchange between the tropical upper water column and the extratropical Southern Ocean enhanced WPWP warming paced by the precession. The WPWP began to substantially expand when La Niña-like conditions peaked at ~12 ka, and the enhanced zonal and meridional thermal gradients further facilitated global warming. Our results underscore the pivotal role of the thermal coupling between the WPWP and the Southern Ocean during the last deglacial warming and its relevance for future global warming.

## Methods

### Compilation of regional SST and SSTA stacks

We integrated 340 published SST records spanning the last 30,000 years from 302 stations in the Pacific and eastern Indian Oceans to reconstruct temporal and spatial distribution and ENSO-like variability since the last glacial. The data set is comprised of 188 planktonic foraminiferal calcite Mg/Ca records based on *G. ruber* and of 152 $U^k_{37}$ records (Fig. 1). Moreover, 7 published subT records based on Mg/Ca measurements of planktonic foraminiferal thermocline species and one new OHC record (Supplementary material) from core KX22-4 (Fig. 1) were incorporated to spatially investigate vertical thermal distribution and to explore the key process involved in global warming during deglaciation.

We directly applied the original SST data considering that the authors have supplied the optimal local SST estimations (Supplementary Data). All of the age models are either radiocarbon-dated or calibrated to the LR04 reference curve[81]. In this study, we specifically focused on the contrast between two key time intervals, i.e., the early Holocene (5–10 ka) and last glacial period (17–29 ka). We considered the last glacial period rather than the LGM (19–23 ka) because the timing of the lowest glacial SST was not consistent in the published records, and the averaged SST for the LGM and last glacial did not differ significantly (0.10 °C, Supplementary Fig. 10). Additionally, in view of the fact that different approaches such as cleaning methods between laboratories, estimations of the dissolution effect would bring offsets to Mg/Ca-based SST reconstructions, we estimated SSTA relative to the mean value of the core top 1 kyr for each downcore SST data.

Cluster analysis was applied to classify the 169 sites in the western Pacific and eastern Indian Ocean areas into four regions (Fig. 1 and Supplementary Fig. 1): the South China Sea, eastern Indian Ocean, west and east part of the WPWP, which is beneficial to explore the spatial thermal transfer between regions. Cluster analysis was performed on SST data (Mg/Ca and $U^k_{37}$) covering both the early Holocene and last glacial periods using software Past 4.08 with Ward's method[82]. Longitude and latitude were taken as references for cluster analysis. Zonal and meridional heat transfer was evaluated by calculating SST differences between regions.

We used STATNARY 1.2 (https://www.marum.de/Prof.-Dr.-michael-schulz/Michael-Schulz-Software.html) to calculate SST/SSTA stacks for all sites within each region with a sliding rectangular window of 1 kyr width (Fig. 2e, and Supplementary Figs. 5c, 12 and 13a, c), which could estimate the time-dependent mean of an unevenly spaced time series directly without the requirement of interpolation. We calculated the SST difference between regions after the data was linearly interpolated to 1 kyr time interval (Fig. 2d, g–i, Supplementary Figs. 5a, b, 7a, 9, and 11b).

### Investigation of WPWP evolution and ENSO mode

We used temporal variations in SST distribution based on 160 SST records from a meridional region (35°N–35°S, 110°E–130°E) and 95 SST records from a zonal region (10°N–10°S, 90°E–180°E) to estimate meridional/zonal shifts of the WPWP during the last 30 kyr. To investigate variations of the WPWP boundaries, data from all sites were linearly interpolated to 1 kyr time interval, orthogonal polynomial fitting (OPF) was applied to data of every 1 kyr SST distribution. We defined the range of the WPWP by the warmest 1 °C based on Mg/Ca-SST trend line through time (Fig. 3), because a uniform criterion is invalid for both glacial and interglacial periods. Then, the positions of the WPWP boundaries and SST maximum could be recognized.

We used SST/SSTA data from sites within the central WPWP (117°E–161°E, 5°S–4°N, modern 29 °C isotherm) to explore the ENSO evolution. The SST variability within the 29 °C isotherm is consistent with that within the 28 °C isotherm but with an offset of 0.45 °C (Supplementary Fig. 11). Then we compared these data with published SST/SSTA data from the cold tongue of EEP (Fig. 1 and Supplementary Table 1), which are commonly used to characterize ENSO variations[12,21], as the large SST anomalies in the cold tongue of EEP are correlated strongly with the Niño-3 index[21].

## Data availability

The data generated in this study have also been deposited in the Pangaea repository (https://doi.pangaea.de/10.1594/PANGAEA.947087). Source data are provided with this paper.

## Code availability

Codes for TRACE are publicly available at https://www.earth systemgrid.org/project/trace.html.

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

## Acknowledgements

This study was supported by the National Natural Science Foundation of China (Grant Nos. 41830539-T.L., 41906058-Z.Y., 42076051-F.C., 41976047-Y.W., and 41876002-X.C.) and Special funds of Shandong Province for Pilot National Laboratory for Marine Science and Technology (Qingdao) (No. 2022QNLM050203-1-T.L.). The authors also gratefully acknowledge the valuable advice from J.Z. and X.W., and appreciate the laboratory assistance of Z.X., H.W., Y.D., F.Q., J.Z., as well as the crews and shipboard scientific parties of cruise KX08-973 and cruise NORC2019-09.

## Author contributions

S.Z., Z.Y., and T.L. designed the study and wrote the manuscript. S.Z. and H.L. contributed to the data analysis and interpretation. Y.W., X.G., A.H., X.C., F.C., and T.L. provided critical comments and revisions.

## Competing interests

The authors declare no competing interests.
