## [Peer Review File · Nature Communications]

Thermal coupling of the Indo-Pacific warm pool and Southern Ocean over the past 30,000 yearsREVIEWER COMMENTS

Reviewer #1 (Remarks to the Author):

The authors utilized published SST records from the West Pacific Warm Pool and EEP along with an additional new ocean heat content record to reconstruct the temperature difference across the Equatorial Pacific over the last ~30 ka. They find that the thermal difference was reduced during the last glacial maximum and increased through deglaciation. They suggest that heat stored in the glacial Southern Ocean was transported to the WPWP, causing the WEP to warm more quickly than the EEP.

I found it a bit difficult to make the connection from your data to the big picture question and mechanisms and would have benefited from a more explanation. Specifically, it would help to have you walk readers through the flow path of the heat from the Southern Ocean to the WEP to more clearly explain why warming more intense in the WEP. As this is geared at a more general audience, it would also help if you explained specifically what you mean when you describe El Nino vs La Nina like conditions.

When reading the main text, I was confused about how the record of ocean heat content was developed and whether it was a foram Mg/Ca record or a compiled record or another proxy etc. It would be helpful to have a brief 1-2 sentence description of the proxy in the main text.

I also would have liked some discussion of how this record is similar to or differs from previous records looking at glacial cross-equatorial SST differences. This manuscript is a compilation of many records, but were the results the same as studies that just used a few records?

Figure 2: The color of El Junco record was too light to see in the figure. Also, there seemed to be a lot of extra information in this plot that was just briefly cited in the text. I think it would help to scale this figure down to just include the essential info. You could eliminate the modeled Nnino3 SSTA (I don't think this record was explained), EEP d13C record, the WDC CO2 and 18O records, and possibly the El Junco % sand record. I also would have liked to have seen the error of the WPWP-EEP Δ SST record plotted.

Figure 4: The EEP and EIO ITF vertical delta T polynomial fitting results seemed weakly related to the raw data. How significant were the relationships for each of the regions in this figure? The letters on the map didn't stand out and were hard to notice, perhaps try making them a darker or bolder blue? It might also help to add a bold title to panel f, something to highlight that it isn't data from a specific region but is a comparison. There was a lot going on in this figure, but I really like the map in the center. It might be more helpful to have a version of this figure that focuses on the spatial/map visualization. It could be a three panel figure with three maps, one showing the modern SSTA, one showing the deglacial as you have here, and one showing the glacial setup. Then, the graphs you have shown in this figure could be a separate figure.

I found section 2 of the supplementary information on the seasonality of temperature records to be very interesting, it might be worth moving this section to the main text along with Figure s10. This has the added benefit of more clearly showing the raw SST stacks.

I also liked figure S3 in the supplementary information and would have liked a version of this figure that was widened to include the EEP.

Line 32: Do you mean insolation?

Helpful Southern Ocean heat content citations:

Pedro, J. B., Martin, T., Steig, E. J., Jochum, M., Park, W., & Rasmussen, S. O. (2016). Southern Ocean deep convection as a driver of Antarctic warming events. *Geophysical Research Letters*, 43(5), 2192–2199. <https://doi.org/10.1002/2016GL067861>
Galbraith, E. D., Merlis, T. M., & Palter, J. B. (2016). Destabilization of glacial climate by the radiative impact of Atlantic Meridional Overturning Circulation disruptions. *Geophysical Research Letters*, 43(15), 8214–8221. <https://doi.org/10.1002/2016GL069846>

Reviewer #2 (Remarks to the Author):

The manuscript “Thermal coupling of the Indo-Pacific warm pool and Southern Ocean over the past 30,000 years” by Zhang Shuai et al presents a compilation of temperature records from across the Indo-Pacific Warm Pool to explore thermal gradients during deglacial warming. The study reconstructs variations in the spatial extent of the WPWP and infers changes in ENSO-like modes by comparing SST anomaly stacks between the WPWP and EEP, meridional and zonal gradients are reconstructed through division of the IPWP into 4 main regions, further the authors also present a new OHC record from core KX22-4.

Despite a range of studies existing that investigate the Pacific mean state via SST/subT reconstructions and compilations across this interval (e.g., Moffa-Sanchez et al, 2019, Dang et al, 2020), the manuscript presents a useful contribution to the literature. I found the manuscript enjoyable to read, comprehensible and references up to date.

However, clarifications and edits are necessary prior to consideration for publication, outlined below.

The overall thrust of the paper is the connection between El Nino and La Nina-like states via SST thermal gradients in the IPWP via the influence of precession and the Southern Ocean. However, I don't think the outline for a Southern Ocean influence on SST evolution is completely strong. There is literature that may be useful to additionally include for linking with the framework of the Southern Ocean influence (e.g. Hwang et al, 2017 Connecting tropical climate change with Southern Ocean heat uptake; Kang et al 2020, Walker circulation response to extratropical radiative forcing). Considering the similarity with Dang et al 2020 (Pacific warm pool subsurface heat sequestration modulated Walker circulation and ENSO activity during the Holocene) I would have liked to have seen more acknowledgement/reference made to the findings of that study as it is not too dissimilar. Particularly, when discussing the implications of thermocline evolution. I'm not entirely convinced by the evidence presented based on Figure 4 to fully support the interpretations of the authors, that is related to the increase in subT prior to SST (4d, e). Further, in this section I do not entirely agree either with the sentence in lines 208-209 as the authors do not explicitly show evidence but is a mechanistic-based interpretation. Consider rewording to be less explicit.

An essential element that I believe to be missing from the manuscript and supplementary materials is a table of the 340 published SST records, providing coordinate information, core name, SST equation originally applied and corresponding reference. Currently, these studies appear to be uncited, and a supplementary table would at least provide some acknowledgement and allow this work to be subsequently built on in future investigations.

In general, I feel some edits are required for the figures. Some of the colour choices/combinations need improved, particularly bearing in mind readers who are colour blind. Figure 1, consider changing the colour of the published SST (green) dots. For example, blue is the best colour to mix with reds and oranges. Figure 2 is quite busy (particularly the upper section), I would suggest either reducing some of the

components of the figure or rearranging for improved clarity. I also am not a fan of the magenta and cyan shaded bars – more neutral colours would perhaps be better here.

SST reconstructions are referred to in Figure 1 as “annual” however, there are inherent seasonal biases attached to the proxies which indeed is discussed in the supplementary materials, and in Line 100 refers to the temperature reconstructions to be seasonally linked.

Minor comments incl. editorial

1. Considering re-phrasing/splitting the sentence in lines 43/44 for improved readability.
2. Reference 14 in line 57 is not relevant reference as study is based on MAT and not geochemical proxy derived SST reconstruction.
3. Figure 1: Include in figure description meaning of the difference in colour of dashed rectangles (i.e. white and black).
4. Line 88: “than the modern annual mean Δ SST (Fig. 2c)” – what do you mean by modern here?
5. Consider reducing the number of abbreviations, for example, I would encourage removing the abbreviations for the Early Holocene (EH) and last glacial period (LG).
6. Figure 2: consider making some alterations to the colouring for improved clarity and reassembling the axes. Be more explicit with “Late Winter Insolation” and “Winter Δ Insolation”
7. When referring to the reconstructed Δ SST e.g., lines 88, 90 and in Table S1 give very definitive errors yet it is unclear how(what) the error (\pm) has been derived, is this calculated in the STATNARY software used to produce the stacks?
8. I would suggest editing the text a little in the following section: Lines 129-130 “A gradually increased frequency of El Nino-like events occurred over the Holocene” then in Lines 133/134 “El Nino-like conditions during the Holocene were not as marked during the LG” may cause some confusion in the reader.
9. Figure 4, I would suggest keeping the panels a-e(f) located next to each other and panel g (map) situated below for ease of comparison. Where possible, it would be ideal to keep the axes ranges the same between panels (e.g., Δ T in panel f). In addition, some panels need the axis range extended to encompass the record fully (e.g., subT record in panel c, SST record in panel b). Inconsistencies are also present in axis titles (e.g., SCP T ($^{\circ}$ C) in panel e versus T ($^{\circ}$ C) in other panels) and colouring, some are coloured while others aren't.
It is also unclear if the Δ T records in panel f are just representative of the differences in SST. Please clarify.
I'm also not entirely sure what is meant by the blue, light red and darker red arrows on panel g, are these oceans currents/hypothesized pathways of heat and if hypothesized do these relate to the expected circulation regime?
The figure legend is missing information related to the dashed SST and subT lines in panel e.
What is the purpose of the black arrows pointing downwards in panels a-e?
10. Supplementary Materials: I would suggest rewriting the Mg/Ca equations used in the typical equation style. For the Anand et al (2003) equation used is this correct?

Response to comments

REVIEWER COMMENTS

Reviewer #1 (Remarks to the Author):

The authors utilized published SST records from the West Pacific Warm Pool and EEP along with an additional new ocean heat content record to reconstruct the temperature difference across the Equatorial Pacific over the last ~30 ka. They find that the thermal difference was reduced during the last glacial maximum and increased through deglaciation. They suggest that heat stored in the glacial Southern Ocean was transported to the WPWP, causing the WEP to warm more quickly than the EEP.

1) I found it a bit difficult to make the connection from your data to the big picture question and mechanisms and would have benefited from a more explanation. Specifically, it would help to have you walk readers through the flow path of the heat from the Southern Ocean to the WEP to more clearly explain why warming more intense in the WEP. As this is geared at a more general audience, it would also help if you explained specifically what you mean when you describe El Nino vs La Nina like conditions.

Agreed. We have followed this constructive comment to strengthen the discussion from the following two aspects: (1) **About the thermal connection between the WEP and Southern Ocean:** via both atmospheric and oceanic processes. ① The current variability in SST, sea-ice extent, and wind stress in the Antarctic region is linked to tropical Pacific ENSO activity via atmospheric teleconnections (White and Peterson, 1996). Many papers have been written about the atmospheric process, and we have presented some of them in lines 152-155. ② We strengthened the oceanic processes by supplementing the literatures and discussions in lines 155-206, which is organized by the statement of circulation, heat transfer and the potential mechanism. In summary, oceanic tunneling transports warm and saltier water from the extratropical Southern Ocean into the Pacific thermocline via the shallow meridional overturning circulation/subtropical overturning cells, which consists of subtropical water subduction, equatorward geostrophic flow into the thermocline to the equatorial Pacific, and upwelling at the equator (central and east Pacific) via equatorial undercurrent (EUC) (Fig.4g). Many observational (Ganachaud et al., 2014; Johnson and McPhaden, 1999; Kolodziejczyk and Gaillard, 2012; Lindstrom et al., 1987; McPhaden and Zhang, 2002; Qu et al., 2008; Zhang and Qu, 2014) and modelling (Luo et al., 2003; Luo et al., 2005; Nonaka and Sasaki, 2007; Qu et al., 2013; Rodgers et al., 2003) studies have confirmed this pathway. Subduction of South Pacific waters include Subtropical Mode Water, Sub-Antarctic Mode Water and Antarctic Intermediate Water (Pena et al., 2008; Qu et al., 2008; Rodgers et al., 2003). As a result, the links between the extratropical and tropical oceans modify the equatorial thermocline and surface equatorial properties, with the potential to modulate ENSO (Ganachaud et al., 2014). The present pattern may shed light on the mechanism of the flow path of the heat from the Southern Ocean to the WPWP during the deglaciation. Thus, we hypothesized that the extratropical warm anomaly has been constantly subducted to the WPWP thermocline and affect ENSO-like changes since 360 ka paced by orbital precession (Zhang et al., 2021). In this manuscript, subT and OHC of WPWP rose at ~22 ka that precede SST warming, which is consistent with two subT records from the South Pacific (Fig. 4) and in phase with precession. In addition, we presented a TRACE result (Liu et al., 2009) that demonstrated that the upper water column of the WPWP

received heat from the south and lower waters (Fig. s8) during deglacial warming. Then we also included ϵ_{Nd} and $\Delta^{14}C$ evidences from the eastern Pacific, which has been fed by EUC since the last glacial period (lines 187-189). To some extent, this process would have contributed to the critical function of teleconnection between the WPWP and Southern Ocean in the deglaciation.

(2) About the more intense warming in the WEP. First, the result of more intense SST warming in the western Pacific and relevant La Niña-like state during the deglacial period should be robust, as detailed in our reply to question #5 from Reviewer #2. The more intense warming in the WEP is then attributed to two processes: the barrier layer, which inhibits vertical mixing and encourages heat built up at the sea surface (Lukas and Lindstrom, 1991; Maes et al., 2002, 2005), and the reduced vertical thermal gradient caused by heat accumulated in the thermocline subducted from the south (discussed above). Because the zonal displacement of heavy precipitation over the western Pacific is an integral aspect of ENSO, the associated barrier layer may play a role. We clarify the deglacial barrier layer in two ways (lines 134-149): we infer its development and effect on deglacial SST warming based on its modern relationship with precipitation, deep convection and warm deep mixed layer in the deglaciation should develop a strong barrier layer in the WPWP. Then, based on our data from the warm pool (core KX22-4), we deduce that the ΔT of *G. ruber* and *T. sacculifer* is 0.5°C, indicating that they both reside in the mixed layer during deglacial warming, but *T. sacculifer* live in waters that are saltier than *G. ruber*, most likely in the barrier layer (Fig. s7). Therefore, both the strong barrier layer and the warm south original thermocline water contributed to the more intense warming in the WEP.

(3) We gave explanations to the ENSO for a general audience, please see lines 41-45. Thanks.

2) When reading the main text, I was confused about how the record of ocean heat content was developed and whether it was a foram Mg/Ca record or a compiled record or another proxy etc. It would be helpful to have a brief 1-2 sentence description of the proxy in the main text.

Agreed. We have added a brief description of it in lines 65-66 in the main text.

3) I also would have liked some discussion of how this record is similar to or differs from previous records looking at glacial cross-equatorial SST differences. This manuscript is a compilation of many records, but were the results the same as studies that just used a few records?

Agreed. We added some detail discussions about it in lines 46-56. Despite a range of studies, the magnitude of glacial SST cooling in the equatorial Pacific varies depending on geochemical proxies and the heterogeneous hydrological conditions in which the sediment cores located, producing 2-5°C cooling in the IPWP and 0.3-3.5°C cooling in the EEP cold tongue. That is why we compiled all of the available data to gain a thorough and precise picture of the thermal change during the deglaciation. Referring to the relevant ENSO-like transition, another issue is that the various approaches employed in ENSO evaluation result in contradictory conclusions concerning the prevalence of La Niña-like or El Niño-like conditions in the last glacial period. We investigated ENSO-like transitions by comparing zonal gradients of SST stacks from WPWP and cold tongue regions to give a robust conclusion.

4) Figure 2: The color of El Junco record was too light to see in the figure. Also, there seemed to be a lot of extra information in this plot that was just briefly cited in the text. I think it would help to scale this figure down to just include the essential info. You could eliminate the modeled Nnino3 SSTA (I don't think this record was explained), EEP d13C record, the WDC CO2 and 18O records, and possibly the El Junco % sand record. I also would have liked to have seen the error of the WPWP-EEP Δ SST record plotted.

Agreed. We removed the unessential information as suggested and added the standard error of the WPWP-EEP Δ SST record in the figure.

5) Figure 4: The EEP and EIO ITF vertical delta T polynomial fitting results seemed weakly related to the raw data. How significant were the relationships for each of the regions in this figure? The letters on the map didn't stand out and were hard to notice, perhaps try making them a darker or bolder blue? It might also help to add a bold title to panel f, something to highlight that it isn't data from a specific region but is a comparison. There was a lot going on in this figure, but I really like the map in the center. It might be more helpful to have a version of this figure that focuses on the spatial/map visualization. It could be a three panel figure with three maps, one showing the modern SSTA, one showing the deglacial as you have here, and one showing the glacial setup. Then, the graphs you have shown in this figure could be a separate figure.

Thanks. The R and P values of the vertical delta T polynomial fitting for each region are provided below, and these values have also been added to Figure 4. The P value confirms that the fitting is significant. In this figure, we are more interested in the deglacial period, and there is a clear trend in all locations. For the EIO, high resolution data in the Holocene appears to be weakly connected to fitting, but there is a clear rising trend in the deglaciation. For the EEP, the data is more scattered, so we tried another method (figure below), in which raw SST and sub-T data were interpolated into 0.2 kyr before calculating the vertical Δ T, and the vertical Δ T showed a clearer trend, while the result of polynomial fitting is identical to the original one in Figure 4c.

Thank you, we have increased the size of the letters in the figure and relocated panel f to Figure 2. Given that it displays horizontal heat transmission, it should be more suited to go alongside Figure 2.

We seriously evaluated the suggestion to display three maps, one for the modern setup and one for the glacial setup. While we would like to concentrate on the process of heat transfer during the deglaciation, which is also the main topic of this manuscript. Thank you again for the good idea.

Region	R value	P value
ECS	0.91	0.0003
EIO	0.79	<0.0001
EEP	0.38	<0.0001
EWP	0.62	<0.0001
SP	0.77	<0.0001

6) I found section 2 of the supplementary information on the seasonality of temperature records to be very interesting, it might be worth moving this section to the main text along with Figure s10. This has the added benefit of more clearly showing the raw SST stacks.

Thank you very much! Some research is attempting to determine the seasonality of Mg/Ca and $U^{K_{37}}$ proxies (Bova et al., 2021; Longo et al., 2020; Timmermann et al., 2014). We did not include this part in the main text for two reasons: (1) it is not the topic of this article and we would like to be more concentrated, and (2) there are still some issues that are unsolved in this text. For example, a recently published research stated that the SST difference between these proxies is due to the models' failure to account for vegetation, but not the seasonality (Thompson et al., 2022). In fact, the ecology of planktonic foraminifera and coccoliths is far more complicated than the expected pattern (Lee et al., 2021a; Lee et al., 2021b; Saher et al., 2009; Sikes et al., 2019; Wang et al., 2013), despite the fact that Mg/Ca and $U^{K_{37}}$ based temperature match the seasonal insolation paced SST in the models. Our findings just confirmed the proxy seasonality hypothesis, with on further improvements. But we do have some opinions about it, which we may share in a future work. Thank you for this excellent suggestion again.

7) I also liked figure S3 in the supplementary information and would have liked a version of this figure that was widened to include the EEP.

Thanks and we have revised it.

8) Line 32: Do you mean insolation?

Yes, it is. We have revised it.

Helpful Southern Ocean heat content citations:

Pedro, J. B., Martin, T., Steig, E. J., Jochum, M., Park, W., & Rasmussen, S. O. (2016). Southern Ocean deep convection as a driver of Antarctic warming events. *GRL*, 43(5), 2192–2199. <https://doi.org/10.1002/2016GL067861>

Galbraith, E. D., Merlis, T. M., & Palter, J. B. (2016). Destabilization of glacial climate by the radiative impact of Atlantic Meridional Overturning Circulation disruptions. *GRL*, 43(15), 8214–8221. <https://doi.org/10.1002/2016GL069846>

Thanks, we carefully read these papers and have benefited greatly from them.

Reviewer #2 (Remarks to the Author):

The manuscript “Thermal coupling of the Indo-Pacific warm pool and Southern Ocean over the past 30,000 years” by Zhang Shuai et al presents a compilation of temperature records from across the Indo-Pacific Warm Pool to explore thermal gradients during deglacial warming. The study reconstructs variations in the spatial extent of the WPWP and infers changes in ENSO-like modes by comparing SST anomaly stacks between the WPWP and EEP, meridional and zonal gradients are reconstructed through division of the IPWP into 4 main regions, further the authors also present a new OHC record from core KX22-4.

Despite a range of studies existing that investigate the Pacific mean state via SST/subT reconstructions and compilations across this interval (e.g., Moffa-Sanchez et al, 2019, Dang et al, 2020), the manuscript presents a useful contribution to the literature. I found the manuscript enjoyable to read, comprehensible and references up to date.

However, clarifications and edits are necessary prior to consideration for publication, outlined below.

1) The overall thrust of the paper is the connection between El Nino and La Nina-like states via SST thermal gradients in the IPWP via the influence of precession and the Southern Ocean. However, I don't think the outline for a Southern Ocean influence on SST evolution is completely strong. There is literature that may be useful to additionally include for linking with the framework of the Southern Ocean influence (e.g. Hwang et al, 2017 Connecting tropical climate change with Southern Ocean heat uptake; Kang et al 2020, Walker circulation response to extratropical radiative forcing).

Thanks. We highly appreciate your insightful comments that helped improve this manuscript. We carefully read the recommended papers and more relevant ones. The two papers are about the atmospheric pathway that we have included them in lines 152 and 155. We enhanced the link between the equatorial Pacific and the Southern Ocean via an oceanic channel in lines 155-206, and please also see our reply to question #1 from Reviewer #1.

2) Considering the similarity with Dang et al 2020 (Pacific warm pool subsurface heat sequestration modulated Walker circulation and ENSO activity during the Holocene) I would have liked to have seen more acknowledgement/reference made to the findings of that study as it is not too dissimilar. Particularly, when discussing the implications of thermocline evolution. I'm not entirely convinced by the evidence presented based on Figure 4 to fully support the interpretations of the authors, that is related to the increase in subT prior to SST (4d, e). Further, in this section I do not entirely agree either with the sentence in lines 208-209 as the authors do not explicitly show evidence but is a mechanistic-based interpretation. Consider rewording to be less explicit.

Agreed. The paper of (Dang et al., 2020) supports us in many ways, and we discussed it in lines 180-181, mainly about the preceding thermocline warming at ~22 ka in their record, ~4000 years prior to the onset of deglacial $p\text{CO}_2$ and SST rise in the IPWP, which we observed to a greater extent in the south and west Pacific. (Dang et al., 2020) focused on the Holocene process and argued that the thermocline warming corresponds to heat transport of southern Pacific shallow overturning

circulation driven by insolation maximum, whereas we focused on the coupling with the Southern Ocean during the deglacial warming.

Some of the mechanisms or explanations in the text are based on modern observations and models that show a clear teleconnection between equatorial Pacific and the Southern Ocean. It is difficult to testify the past using proxy data, but we attempted to do so in (Zhang et al., 2021), and warm anomaly was strengthened and transported to equatorial thermocline in precession minimum in the models. In this manuscript, we included several evidences of ϵ_{Nd} and $\Delta^{14}C$ records from the eastern Pacific, which has been fed by EUC since the last glacial period, as well as our data on the thermal transfer process and TRACE results (please see our reply to question #1 from Reviewer #1 and lines 155-206 in the manuscript). To some extent, we argued for a possible teleconnection between the WPWP and Southern Ocean during the deglaciation.

We removed the improper statement and changed to a less explicit way (lines 184-185).

3) An essential element that I believe to be missing from the manuscript and supplementary materials is a table of the 340 published SST records, providing coordinate information, core name, SST equation originally applied and corresponding reference. Currently, these studies appear to be uncited, and a supplementary table would at least provide some acknowledgement and allow this work to be subsequently built on in future investigations.

Agreed. We had added a supplementary data file with the information of the sites. We also submit our data to the PANGAEA website (PDI-32274).

4) In general, I feel some edits are required for the figures. Some of the colour choices/combinations need improved, particularly bearing in mind readers who are colour blind. Figure 1, consider changing the colour of the published SST (green) dots. For example, blue is the best colour to mix with reds and oranges. Figure 2 is quite busy (particularly the upper section), I would suggest either reducing some of the components of the figure or rearranging for improved clarity. I also am not a fan of the magenta and cyan shaded bars – more neutral colours would perhaps be better here.

Agreed. The color of the figures has been enhanced. We moved the upper section of Figure 2 to Figure 3, which tells the story of the warm pool's evolution. Then, we reduced some components following the comment #4 from the Reviewer #1 and move Figure 4f into Figure 2, which we believe is more reasonable. We changed the color of the bar in all of the pictures.

5) SST reconstructions are referred to in Figure 1 as “annual” however, there are inherent seasonal biases attached to the proxies which indeed is discussed in the supplementary materials, and in Line 100 refers to the temperature reconstructions to be seasonally linked.

Agreed. We have carefully considered about it. There is only one $U^{K'}_{37}$ record for ENSO-like investigation within the WPWP region, which is based on core MD97-2138 (146.14°E, 1.25°N, (de Garidel-Thoron et al., 2007)). However, the LGM alkenone-based temperature is $\sim 2^\circ\text{C}$ higher than Mg/Ca-T, and a lesser 1.3°C cooling was seen in LGM than during the Holocene, and the authors thought the $U^{K'}_{37}$ could reflect subsurface temperature. Besides, three other $U^{K'}_{37}$ records that were beyond the range of 5°S - 4°N in the warm pool had a similar character (Fig. s12). Given another issue with the limit of 28.2°C for the $U^{K'}_{37}$ proxy, we think it is risky to include the only one $U^{K'}_{37}$ data from core MD97-2138 in the ENSO-like mode investigation. Of course, we also considered that

seasonality in the WPWP is fairly low, around 0.67°C . Overall, we think the WPWP Mg/Ca-T may accurately reflect annual temperature, or that the seasonality inaccuracy is limited.

Although we cannot confirm whether or not/how much WPWP Mg/Ca-T is seasonally connected. The result of more intense deglacial warming in the WPWP and the concomitant La Niña-like state is robust, because the LG-EH ΔSST based on WPWP Mg/Ca is greater than both Mg/Ca and U^{K}_{37} , as well as annual temperature in the EEP (Tab. s1). Therefore, the ENSO evolution based on the zonal gradient between WPWP Mg/Ca and EEP Mg/Ca/ U^{K}_{37} /annual all yields similar trends and conclusions (Fig. s5).

Minor comments incl. editorial

1. Considering re-phrasing/splitting the sentence in lines 43/44 for improved readability.

Agreed. We have carefully rephrased the expressions, please see the first paragraph of the introduction.

2. Reference 14 in line 57 is not relevant reference as study is based on MAT and not geochemical proxy derived SST reconstruction.

Thanks, our apology for the mistake. We deleted it and rephrased the entire paragraph.

3. Figure 1: Include in figure description meaning of the difference in colour of dashed rectangles (i.e. white and black).

Agreed. We have explained it in line 75.

4. Line 88: “than the modern annual mean ΔSST (Fig. 2c)” – what do you mean by modern here?

Thanks. We added the modern line in Fig. 2d for reference, which is based on the WOA18 SST data from the sites (specify it in line 96).

5. Consider reducing the number of abbreviations, for example, I would encourage removing the abbreviations for the Early Holocene (EH) and last glacial period (LG).

Agreed. We have spell out EH and LG in the entire text.

6. Figure 2: consider making some alterations to the colouring for improved clarity and reassembling the axes. Be more explicit with “Late Winter Insolation” and “Winter Δ Insolation”

Agreed. We have deleted the Winter Δ Insolation because we adjusted the text.

7. When referring to the reconstructed ΔSST e.g., lines 88, 90 and in Table S1 give very definitive errors yet it is unclear how(what) the error (\pm) has been derived, is this calculated in the STATNARY software used to produce the stacks?

Thanks. The errors were SD for the raw data of $\Delta\text{SST}_{\text{LG-EH}}$ of all the sites within each region, but not the error calculated by the STATNARY software.

8. I would suggest editing the text a little in the following section: Lines 129-130 “A gradually increased frequency of El Niño-like events occurred over the Holocene” then in Lines 133/134 “El Niño-like conditions during the Holocene were not as marked during the LG” may cause some

confusion in the reader.

Agreed. We have carefully rephrased the expressions.

9. Figure 4, I would suggest keeping the panels a-e(f) located next to each other and panel g (map) situated below for ease of comparison. Where possible, it would be ideal to keep the axes ranges the same between panels (e.g., ΔT in panel f). In addition, some panels need the axis range extended to encompass the record fully (e.g., subT record in panel c, SST record in panel b). Inconsistencies are also present in axis titles (e.g., SCP T ($^{\circ}\text{C}$) in panel e versus T ($^{\circ}\text{C}$) in other panels) and colouring, some are coloured while others aren't.

It is also unclear if the ΔT records in panel f are just representative of the differences in SST. Please clarify.

I'm also not entirely sure what is meant by the blue, light red and darker red arrows on panel g, are these oceans currents/hypothesized pathways of heat and if hypothesized do these relate to the expected circulation regime?

The figure legend is missing information related to the dashed SST and subT lines in panel e.

What is the purpose of the black arrows pointing downwards in panels a-e?

Agreed. We have revised the figure as request. Thank you for your precious comments.

The ΔT records in panel f are representative of the differences in SST, and we have moved it to Figure 2 that we think it is more reasonable for both Figure 2 and 4.

The arrows on panel g are hypothesized pathways of heat and we have clarified in the figure legend. The hypothesized pathways are related to the expected circulation regime, and we have clarified it in the legend.

Our apology for any missing information. The solid and dashed lines represented two cores used in the south Pacific, marked as locations "e" and "f" in Figure 4g. We now separated them into Figs. 4e and 4f because they are located far apart and have a large temperature difference.

We specified the meaning of arrow in the figure legend.

10. Supplementary Materials: I would suggest rewriting the Mg/Ca equations used in the typical equation style. For the Anand et al (2003) equation used is this correct?

Agreed. We have revised it.

References:

- Bova, S., Rosenthal, Y., Liu, Z., Godad, S.P., Yan, M., 2021. Seasonal origin of the thermal maxima at the Holocene and the last interglacial. *Nature* 589, 548-553.
- Dang, H., Jian, Z., Wang, Y., Mohtadi, M., Rosenthal, Y., Ye, L., Bassinot, F., Kuhnt, W., 2020. Pacific warm pool subsurface heat sequestration modulated Walker circulation and ENSO activity during the Holocene. *Science Advances* 6, eabc0402.
- de Garidel-Thoron, T., Rosenthal, Y., Beaufort, L., Bard, E., Sonzogni, C., Mix, A.C., 2007. A multiproxy assessment of the western equatorial Pacific hydrography during the last 30 kyr. *Paleoceanography* 22, PA3204.
- Ganachaud, A., Cravatte, S., Melet, A., Schiller, A., Holbrook, N.J., Sloyan, B.M., Widlansky, M.J.,

- Bowen, M., Verron, J., Wiles, P., Ridgway, K., Sutton, P., Sprintall, J., Steinberg, C., Brassington, G., Cai, W., Davis, R., Gasparin, F., Gourdeau, L., Hasegawa, T., Kessler, W., Maes, C., Takahashi, K., Richards, K.J., Send, U., 2014. The southwest Pacific ocean circulation and climate experiment (SPICE). *J. Geophys. Res.: Oceans* 119, 7660-7686.
- Johnson, G.C., McPhaden, M.J., 1999. Interior Pycnocline Flow from the Subtropical to the Equatorial Pacific Ocean. *J. Phys. Oceanogr.* 29, 3073-3089.
- Kolodziejczyk, N., Gaillard, F., 2012. Observation of spiciness interannual variability in the Pacific pycnocline. *J. Geophys. Res.: Oceans* 117.
- Lee, K.E., Clemens, S.C., Kubota, Y., Timmermann, A., Holbourn, A., Yeh, S.-W., Bae, S.W., Ko, T.W., 2021a. Roles of insolation forcing and CO₂ forcing on late Pleistocene seasonal sea surface temperatures. *Nature Communications* 12, 5742.
- Lee, K.E., Park, W., Yeh, S.-W., Bae, S.W., Ko, T.W., Lohmann, G., Nam, S.-I., 2021b. Enhanced climate variability during the last millennium recorded in alkenone sea surface temperatures of the northwest Pacific margin. *Global and Planetary Change* 204, 103558.
- Lindstrom, E., Lukas, R., Fine, R., Firing, E., Godfrey, S., Meyers, G., Tsuchiya, M., 1987. The Western Equatorial Pacific Ocean Circulation Study. *Nature* 330, 533-537.
- Liu, Z., Otto-Bliesner, B.L., He, F., Brady, E.C., Tomas, R., Clark, P.U., Carlson, A.E., Lynch-Stieglitz, J., Curry, W., Brook, E., Erickson, D., Jacob, R., Kutzbach, J., Cheng, J., 2009. Transient simulation of last deglaciation with a new mechanism for Bølling-Allerød warming. *Science* 325, 310-314.
- Longo, W.M., Huang, Y., Russell, J.M., Morrill, C., Daniels, W.C., Giblin, A.E., Crowther, J., 2020. Insolation and greenhouse gases drove Holocene winter and spring warming in Arctic Alaska. *Quat. Sci. Rev.* 242, 106438.
- Lukas, R., Lindstrom, E., 1991. The mixed layer of the western equatorial Pacific ocean. *J. Geophys. Res.: Oceans* 96, 3343-3357.
- Luo, J.-J., Masson, S., Behera, S., Delecluse, P., Gualdi, S., Navarra, A., Yamagata, T., 2003. South Pacific origin of the decadal ENSO-like variation as simulated by a coupled GCM. *Geophys. Res. Lett.* 30, 2250.
- Luo, Y., Rothstein, L.M., Zhang, R.-H., Busalacchi, A.J., 2005. On the connection between South Pacific subtropical spiciness anomalies and decadal equatorial variability in an ocean general circulation model. *J. Geophys. Res.: Oceans* 110, C10002.
- Maes, C., Picaut, J., Belamari, S., 2002. Salinity barrier layer and onset of El Niño in a Pacific coupled model. *Geophys. Res. Lett.* 29, 2206.
- Maes, C., Picaut, J., Belamari, S., 2005. Importance of the salinity barrier layer for the buildup of El Niño. *J. Clim.* 18, 104-118.
- McPhaden, M.J., Zhang, D., 2002. Slowdown of the meridional overturning circulation in the upper Pacific Ocean. *Nature* 415, 603-608.
- Nonaka, M., Sasaki, H., 2007. Formation Mechanism for Isopycnal Temperature–Salinity Anomalies Propagating from the Eastern South Pacific to the Equatorial Region. *J. Clim.* 20, 1305-1315.
- Pena, L.D., Cacho, I., Ferretti, P., Hall, M.A., 2008. El Niño–Southern Oscillation–like variability during glacial terminations and interlatitudinal teleconnections. *Paleoceanography* 23, PA3101.
- Qu, T., Gao, S., Fine, R.A., 2013. Subduction of south Pacific tropical water and its equatorward pathways as shown by a simulated passive tracer. *J. Phys. Oceanogr.* 43, 1551-1565.
- Qu, T., Gao, S., Fukumori, I., Fine, R.A., Lindstrom, E.J., 2008. Subduction of south Pacific waters. *Geophys. Res. Lett.* 35, L02610.

- Rodgers, K.B., Blanke, B., Madec, G., Aumont, O., Ciais, P., Dutay, J.-C., 2003. Extratropical sources of equatorial Pacific upwelling in an OGCM. *Geophys. Res. Lett.* 30, 1084.
- Saher, M.H., Rostek, F., Jung, S.J.A., Bard, E., Schneider, R.R., Greaves, M., Ganssen, G.M., Elderfield, H., Kroon, D., 2009. Western Arabian Sea SST during the penultimate interglacial: A comparison of $U^{K'}_{37}$ and Mg/Ca paleothermometry. *Paleoceanography* 24.
- Sikes, E.L., Schiraldi Jr, B., Williams, A., 2019. Seasonal and Latitudinal Response of New Zealand Sea Surface Temperature to Warming Climate Since the Last Glaciation: Comparing Alkenones to Mg/Ca Foraminiferal Reconstructions. *Paleoceanogr. Paleoclimatol.* 34, 1816-1832.
- Thompson, A., J., Zhu, J., Poulsen, C., J., Tierney, J., E., Skinner, C., B., 2022. Northern Hemisphere vegetation change drives a Holocene thermal maximum. *Science Advances* 8, eabj6535.
- Timmermann, A., Sachs, J., Timm, O.E., 2014. Assessing divergent SST behavior during the last 21 ka derived from alkenones and *G. ruber*-Mg/Ca in the equatorial Pacific. *Paleoceanography* 29, 680-696.
- Wang, Y.V., Leduc, G., Regenberg, M., Andersen, N., Larsen, T., Blanz, T., Schneider, R.R., 2013. Northern and southern hemisphere controls on seasonal sea surface temperatures in the Indian Ocean during the last deglaciation. *Paleoceanography* 28, 2013PA002458.
- White, W.B., Peterson, R.G., 1996. An Antarctic circumpolar wave in surface pressure, wind, temperature and sea-ice extent. *Nature* 380, 699-702.
- Zhang, L., Qu, T., 2014. Low-frequency variability of South Pacific Tropical Water from Argo. *Geophys. Res. Lett.* 41, 2441-2446.
- Zhang, S., Yu, Z., Gong, X., Wang, Y., Chang, F., Lohmman, G., Qi, Y., Li, T., 2021. Precession cycles of the El Niño/Southern Oscillation-like system controlled by Pacific upper-ocean stratification. *Communications Earth & Environment* 2, 239.

REVIEWERS' COMMENTS

Reviewer #1 (Remarks to the Author):

The authors have sufficiently addressed all of my previous comments. I look forward to seeing their future work and hearing their opinions on proxy seasonality!

I have one very minor comment, Lines 187-189: I would also leave out the Pena et al. (2013) reference, later studies suggest that cleaned planktic foraminifera document a bottom water rather than surface/thermocline signal (Kraft et al., 2013; Tachikawa et al., 2014) and the EEP in particular is a tricky region for Nd (Grasse et al., 2012, 2017). Instead, perhaps include reference to Martínez-Fontaine et al. (2019) for eastern South Pacific 14C (a downstream EUC influence) and/or to Umling et al. (2017) for EEP EUC 14C.

1. Grasse, P. et al. Short-term variability of dissolved rare earth elements and neodymium isotopes in the entire water column of the Panama Basin. *Earth and Planetary Science Letters* 475, 242–253 (2017).
2. Grasse, P. et al. Short-term variability of dissolved rare earth elements and neodymium isotopes in the entire water column of the Panama Basin. *Earth and Planetary Science Letters* 475, 242–253 (2017).
3. Tachikawa, K., Piotrowski, A. M. & Bayon, G. Neodymium associated with foraminiferal carbonate as a recorder of seawater isotopic signatures. *Quaternary Science Reviews* 88, 1–13 (2014).
4. Kraft, S., Frank, M., Hathorne, E. C. & Weldeab, S. Assessment of seawater Nd isotope signatures extracted from foraminiferal shells and authigenic phases of Gulf of Guinea sediments. *Geochimica et Cosmochimica Acta* 121, 414–435 (2013).
5. Martínez Fontaine, C. et al. Ventilation of the Deep Ocean Carbon Reservoir During the Last Deglaciation. *Paleoceanography and Paleoclimatology* 34, 2080–2097 (2019).
6. Umling, N. E. & Thunell, R. C. Synchronous deglacial thermocline and deep-water ventilation in the eastern equatorial Pacific. *Nat. Commun.* 8, 14203 (2017).

Reviewer #2 (Remarks to the Author):

I am satisfied with the responses and subsequent changes/alterations made by Zhoufei et al in relation to the manuscript "Thermal coupling of the Indo-Pacific warm pool and Southern Ocean over the past 30,000 years" following review.

Minor remaining comments (mainly in relation to addition of barrier layer):

1. Figure 1 is missing in manuscript merged file.
2. Line 134 consider rephrasing "here, we try to introduce a modern concept of "barrier layer"..." for example, alternative wording "The more intense deglacial warming in the WPWP may be attributed to the presence of a barrier layer, a feature well-recognised in modern observations and models.."
3. Line 140 consider rephrasing "going back in time, La Niña-like state" to be more explicit with what is meant here temporally.
4. Sentence in lines 146-147 needs clarification/re-wording for improved clarity.

Finally, I agree with the proposed mechanism of the barrier layer (and is illustrated nicely by the difference in the $d_{18}O_{sw}$ between the two mixed layer species) however, I believe improved clarity for the reader may be required. Mainly with respect to Line 141 where barrier layer is ascribed to be caused by the La Niña-like state whilst in the conclusions (line 235/236) the La Niña-like state is ascribed to being caused by the barrier layer.

Response to comments

REVIEWERS' COMMENTS

Reviewer #1 (Remarks to the Author):

The authors have sufficiently addressed all of my previous comments. I look forward to seeing their future work and hearing their opinions on proxy seasonality!

Thank you very much!

I have one very minor comment, Lines 187-189: I would also leave out the Pena et al. (2013) reference, later studies suggest that cleaned planktic foraminifera document a bottom water rather than surface/thermocline signal (Kraft et al., 2013; Tachikawa et al., 2014) and the EEP in particular is a tricky region for Nd (Grasse et al., 2012, 2017). Instead, perhaps include reference to Martinez-Fontaine et al. (2019) for eastern South Pacific 14C (a downstream EUC influence) and/or to Umling et al. (2017) for EEP EUC 14C.

Agreed. We carefully read these papers and benefited greatly from them. We have replaced Pena et al. (2013) with Umling et al. (2017).

1. Grasse, P. et al. Short-term variability of dissolved rare earth elements and neodymium isotopes in the entire water column of the Panama Basin. *Earth and Planetary Science Letters* 475, 242–253 (2017).
2. Grasse, P. et al. Short-term variability of dissolved rare earth elements and neodymium isotopes in the entire water column of the Panama Basin. *Earth and Planetary Science Letters* 475, 242–253 (2017).
3. Tachikawa, K., Piotrowski, A. M. & Bayon, G. Neodymium associated with foraminiferal carbonate as a recorder of seawater isotopic signatures. *Quaternary Science Reviews* 88, 1–13 (2014).
4. Kraft, S., Frank, M., Hathorne, E. C. & Weldeab, S. Assessment of seawater Nd isotope signatures extracted from foraminiferal shells and authigenic phases of Gulf of Guinea sediments. *Geochimica et Cosmochimica Acta* 121, 414–435 (2013).
5. Martínez Fontaine, C. et al. Ventilation of the Deep Ocean Carbon Reservoir During the Last Deglaciation. *Paleoceanography and Paleoclimatology* 34, 2080–2097 (2019).
6. Umling, N. E. & Thunell, R. C. Synchronous deglacial thermocline and deep-water ventilation in the eastern equatorial Pacific. *Nat. Commun.* 8, 14203 (2017).

Reviewer #2 (Remarks to the Author):

I am satisfied with the responses and subsequent changes/alterations made by Zhoufei et al in relation to the manuscript "Thermal coupling of the Indo-Pacific warm pool and Southern Ocean over the past 30,000 years" following review.

Thank you very much!

Minor remaining comments (mainly in relation to addition of barrier layer):

1. Figure 1 is missing in manuscript merged file.

Our apology for the mistake, we will check it very carefully before the final submission.

2. Line 134 consider rephrasing "here, we try to introduce a modern concept of "barrier layer"..." for example, alternative wording "The more intense deglacial warming in the WPWP may be attributed to the presence of a barrier layer, a feature well-recognised in modern observations and models.."

Agreed. We have revised it (lines 113-114).

3. Line 140 consider rephrasing "going back in time, La Niña-like state" to be more explicit with what is meant here temporally.

Agreed. We have rephrased the sentence to start with "During the deglaciation..." to clarify the time interval we meant here (lines 118).

4. Sentence in lines 146-147 needs clarification/re-wording for improved clarity.

Agreed. We have rephrased the sentences in lines 123-126.

Finally, I agree with the proposed mechanism of the barrier layer (and is illustrated nicely by the difference in the $d_{18}O_{sw}$ between the two mixed layer species) however, I believe improved clarity for the reader may be required. Mainly with respect to Line 141 where barrier layer is ascribed to be caused by the La Niña-like state whilst in the conclusions (line 235/236) the La Niña-like state is ascribed to being caused by the barrier layer.

Agreed. We have revised the improper statement in lines 119-120.